# Estimation of Effects of Recent Macroprudential Policies in a Sample of Advanced Open Economies

Ragnar Nymoen [1], Kari Pedersen [2] and Jon Ivar Sjåberg [2,*]

1   Department of Economics, University of Oslo, 0315 Oslo, Norway; ragnar.nymoen@econ.uio.no
2   The Financial Supervisory Authority of Norway, 0107 Oslo, Norway; Kari.Pedersen@finanstilsynet.no
*   Correspondence: Jon.Ivar.Sjaberg@finanstilsynet.no

**Abstract:** We used a time-series cross-section dataset to test several hypotheses pertaining to the role of macroprudential policy instruments in the management of the financial cycle in advanced open economies. The short-run effects are most significant for caps on loan to value and income (LTV and LTI) and risk weights (RW). The long-run coefficients of credit growth with respect to the indicators of amortisation requirements (Amort) and RW are also significant. The estimation results when house price growth is the dependent variable are consistent with these results. Our findings do not support that Basel III type countercyclical buffer (CCyB) has affected credit growth, and we suggest that the variable is mainly a control in our dataset. In that interpretation, it is interesting that the estimated coefficients of the other instruments are robust with respect to exclusion of CCyB from the empirical models. The main results are also robust to controls in the form of impulse indicator saturation (IIS), which we employed as a novel estimation method for macro panels.

**Keywords:** macroprudential policy measures; house prices; credit growth; open economies; cross-section time-series; macro panel; impulse indicator saturation; robust estimation

**JEL Classification:** C23; C44; C58; G28; G38

## 1. Introduction

In this paper, we estimate the effects of macroprudential policies on credit growth and housing price changes. The importance of credit and debt for large macroeconomic fluctuations has long been recognised. Irving Fisher's theory of asset price booms and debt-deflation depressions remains an early milestone, and it is interesting to note that, during the first decades after the World War II, leading economists were analysing the increased risk of crisis if and when credit markets again became liberalised (see Anundsen et al. (2014)). In the late 1980s and the early 1990s, boom–busts occurred in Norway, Finland, Sweden and the UK (see Eika and Nymoen (1992), Honkapoja et al. (1996), Honkapoja (2009), and Muellbauer and Murphy (1990)). In the case of Norway, empirical econometric modelling results indicated clearly that both the house price boom, and the ensuing bust, contributed to GDP expansion and recession trough the private saving channel (cf. Brodin and Nymoen (1992) and Eitrheim et al. (2002)).

In recent years, in particular following the global financial crisis that became an income and job crisis, several empirical contributions have recognised the importance of debt and of housing prices in the complex processes that drive economic fluctuations in advanced economies with liberalised credit markets. For example, three studies with consistent findings for the US saving rate dynamics were presented by Carroll et al. (2012), Mian et al. (2013) and Anundsen and Nymoen (2019). Using a dataset with long historical time series for 14 advanced economies, Jordà et al. (2013) concluded that financial factors play an important role in the modern business cycle (see also Anundsen et al. (2016)).

There is a growing consensus about financial crises becoming more serious when credit and house prices have grown sharply in the build-up to the crisis, which it has become custom to include in the calibration of macroprudential stress tests (Anderson et al. (2018)).

Macroprudential policy aims at promoting financial stability partly by, e.g., managing growth in asset prices and credit. Excess growth in these variables over extended periods may be seen as a necessary condition for financial instability (see Borio and Lowe (2002), Reinhardt and Rogoff (2009), Schularick and Taylor (2012), Fahr and Fell (2017) among others). Macroprudential policies however represent a wide menu of policy instruments which has varied in operationalisation both through time and between countries. For example, reserve requirement ratios and limits on foreign currency loans tend to be more used by emerging countries, while caps on loan to value (LTV) have been more used by advanced economies, notably after the financial crisis (see Cerutti et al. (2016)).

To reduce unwanted cross-section heterogeneity, we analysed a time-series cross-section (TSCS) panel dataset consisting of ten advanced and open economies, which are considered to be relatively homogenous in their use of policy measures. The studies that have used (wide) macro panels also indicate that cross section heterogeneity is a factor to be reckoned with. Specifically, the relationship between financial development and macroeconomic activity appears to be stronger in the more advanced open economies (see Chortaes et al. (2015)). This is one reason economies can differ substantially in their response to the various types of policies. In addition to heterogeneity in response, there may be differences in instrument design and operationalisation, meaning that there may be relatively few instances of a given type of instrument even if the cross section dimension is large. Hence, robustness to inclusion of individual countries need careful consideration Kuttner and Shim (2016).

On the other hand, heterogeneity can also give rise to new hypotheses to be tested. For example, Akinci and Olmstead-Rumsey (2017) found evidence that policies directed toward the housing market are driving the results for advanced economies, while non-housing measures matter more for countries that represent emerging markets, and Cerutti et al. (2016) found results supporting that macroprudential measures are less effective for open economies than for relatively closed economies.

The policy measures that we included represent a relatively homogenous set of policy measures in the 10-country sample. The measures are caps (i.e., limits) on loan to value ratios (LTV) and on loan to income ratios (LTI). We also included tools of the type debt service to income (DSTI), risk weights (i.e., in capital requirement regulations of credit institutions, RW) and amortisation requirements (Amort). Again, the idea was that these measures are likely to be sufficiently country-independent to allow time variation in one of the countries to be used (by estimation) to learn about the hypothetical effect of its use in the other countries in the sample.

As a new contribution to the empirical literature on macroprudential policy effects, we included in the panel dataset the level of countercyclical buffer rates (CCyB), as proposed by the Basel Committee on Banking Supervision BCBS (2010a). According to the Basel Committee's multi country study, stronger capital and liquidity buffers can lead to reduced credit, with negative short-term consequences for the economic activity level BCBS (2010b). In modern economies, there is reason to expect that any wide reaching effects of stronger buffer requirements will involve housing markets and prices, and we therefore tested such a hypothesis within the larger model that also takes account of the standard macroprudential measures.

We contribute to the methodology of macroprudential policy effectiveness testing along several dimensions. We note that the custom in the literature, which is to have real credit growth and real house price change as dependent variables, may introduce measurement errors relative to the variable that the policy maker naturally target, which is nominal credit growth specifically. While this measurement error does not necessarily bias the estimates, i.e., if care is taken to use a well specified econometric model equation, it is nevertheless an unnecessary complication that may imply larger than optimal estimated coefficient standard errors. Hence, our main results are for nominal growth rates. However, we also took the custom of the literature into consideration by presenting sensitivity analysis where, e.g., real credit growth is the dependent variable.

Another methodological contribution is that we employed a robust estimator that controls for location breaks in the distribution of the dependent variables. This method was developed for empirical macroeconometric modelling, but has been demonstrated to be useful as a robust estimator of effects of changes in labour market institutions in macro panels (cf. Johansen and Nielsen (2009) and Nymoen and Sparrman (2015)).

In the literature that makes use of macro panels, there is a tendency of including the macroprudential policy measures sequentially, one at a time, to a baseline estimation equation. When the hypotheses are about effectiveness of individual instruments, the obvious problem is that the estimated coefficients estimate joint effects, rather than of intended individual effects (i.e., relative to the specified information set). In our approach, we always included all variables jointly, thus avoiding omitted variable bias and chance positives due to sequential estimation of the effects of macro prudential policy variables.

A final methodological point that we make is that, even though credit growth, housing prices and macroprudential policies are endogenous in a large and complex system, the coefficients of conditional models can nevertheless be relevant parameters of interest when the purpose of the research exercise is the empirical investigation of tangible effects of macroprudential policies.

The remainder of the paper is organised as follows: Section 2 overviews the empirical literature on macroprudential policies most directly related to our own investigation. Section 3 motivates and explains the dataset used, and, in Section 4 several methodological issues relating to model formulation and estimation methods are discussed. Section 5 presents the empirical results for the hypotheses we test: about joint and individual policy instrument significance, differentiation between impact and long-run coefficients, and between credit and housing price effects, as well as several sensitivity tests. Finally, we test the hypothesis that the new Basel III type countercyclical buffer measures has affected credit growth and/or house price change in our sample of advanced open economies. Section 6 concludes.

## 2. The Empirical Literature

During the fist decade of the new millennium, the field of macroprudential policy has come of age (see, e.g., Galati and Moesner (2013) and Anderson et al. (2018)). Already, a sizeable literature has considered the impact and medium term effects of macroprudential policy instruments on measures of credit growth and house price change.

In an important macro panel data study, Kuttner and Shim (2016) found evidence that introduction or reductions of the maximum debt-service-to-income (DSTI) and changes in maximum LTV ratios have been effective for real credit growth. They found that typical DSTI tightening lowered the real credit growth rate by 4–6 per cent over four quarters. The estimation results were considerably weaker for real house prices, both numerically and in terms of statistical significance. The dataset analysed by Kuttner and Shim (2016) consists of 55 countries and quarterly data for the period 1980–2012. The same overall picture emerges from the results of Cerutti et al. (2017), who had an even broader cross-section (119 countries) but much shorter time series dimension (2001–2013). Interestingly, they found that effects are smaller in financially advanced and open economies. Specifically, they mentioned that macroprudential polices may be undermined (circumvented) by cross border banking and other forms of external financing, a reminder that country heterogeneity is a concern.

Stronger evidence for effects on housing prices was found by Akinci and Olmstead-Rumsey (2017), who focused on the imbalances in the housing sector. Carresas et al. (2018) extended the econometric framework used in the literature to cointegration, and found that macroprudential policies can curb house price growth and credit growth also in that econometric model. Dell'Ariccia et al. (2012) presented econometric results which indicate that macroprudential policies can reduce the incidence of credit booms, and reduce the risk of credit booms going wrong (i.e., becoming a banking crisis). In an innovative econometric study, Richter et al. (2018) found that maximum LTV ratios have a duality

about them: They affect credit growth and house prices substantially but have relatively modest effects on output and inflation.

Claessens et al. (2013) investigated how changes in maximum LTV and DSTI rates affect individual banks, and found that both asset and leverage growth are reduced. Gambacorta (2017) used registry data from Latin American countries, Canada and the USA. One of their main findings is that macroprudential policies have been quite effective in stabilising credit cycles. In this study, the macroprudential tools have a greater effect on credit growth when reinforced by the use of conventional monetary policy to push in the same direction. In a study of UK banks, Aiyar et al. (2014) found that variation in bank capital requirements during the 1990s and early 2000s became somewhat ineffective due to the lending strategies of branches of foreign banks. It is plausible that similar constellations exist in other advanced open economies, which go some way in explaining the finding about macroprudential policies being less forceful in open than in more closed modern economies.

Compared to the other main macroprudential policy instruments, the literature contains less empirical analysis of changes in countercyclical buffer rates (CCyB), although that is likely to change as central banks develop their operational frameworks for macroprudential stress testing (see Andersen et al. (2019) and the references therein). The work of Akram (2014) is a notable exception. He investigated macro effects of higher bank capital requirements by extending an empirical macroeconometric model for the Norwegian economy (Bårdsen et al. (2005)) and found that increases in capital requirements were primarily transmitted via lending rates to the other variables of the model. The model simulations indicate that, under the recommendations of Basel III, increased buffer rates would have significant effects on house prices and credit, but weaker than in the results reported by BCBS (2010b).

In the studies that make use of macro panels, and which therefore are comparable to our own investigation, there is a tendency of including the macroprudential policy measures sequentially, one at a time, to a baseline estimation equation or constructing a composite indicator and to estimate the effect of that joint macroprudential policy variable. In both practices, the estimated coefficients are interpretable as estimates of joint effects, rather than of individual effects (i.e., relative to the specified information set). Notable exceptions are Kuttner and Shim (2016) and Akinci and Olmstead-Rumsey (2017), who reported results also for model equations where the coefficients of LTV, DSTI and risk weights (RW) are estimated jointly.

In our approach, we always included all variables jointly, thus avoiding omitted variable bias and chance positives due to sequential estimation of the effects of macroprudential policy variables. In several other respects, our approach inherits a lot from Kuttner and Shim (2016), although the list of macroprudential instruments only partly overlap theirs (specifically, in terms of LTV, DSTI and RW). This is a consequence of our choice of a using a relatively homogenous sample, both in terms of economy characteristics and of policies that have been implemented. The motivation was that, with such a sample, the concern that one otherwise has about panel data regression, i.e., the results might be unduly influenced by the inclusion of a single county, is reduced, at least to some extent.

## 3. The Dataset

Our analysis used data from 10 advanced economies that have implemented macroprudential instruments to counter unsustainable household indebtedness and housing price growth since 1998. The instruments included in the analysis are: (i) limits to Loan-to-Value (LTV); (ii) limits to Loan-to-Income (LTI); (iii) limits to Debt-to-Income (DTI); (iv) limits to Debt-Service-to-Income (DSTI); (v) Amortisation requirements (Amort); (vi) tightening of Risk Weights on mortgage loans in the capital adequacy regulation of credit institutions (RW); and (vii) level of Countercyclical Buffer rate (CCyB). To simplify the smorgasbord of instruments in the analysis, DTI was merged in LTI,

since these instruments are very similar.[1] Another simplification lies in the definition of DSTI. Here, DSTI included both regulation on the share of income spent on debt servicing (i.e., interests paid, instalments and fees), and imposing on the credit institutions an obligation to carry out debt servicing stress tests to assure that the borrowers are capable to service their mortgages also in the case of a certain interest rate increase.

As mentioned in the Introduction, as a novel trait, the countercyclical buffer rate (CCyB) was included in the set of instruments, even though it is not targeted to impact mortgage loans specifically. However, some claim that, due to increased financing costs, build up of CCyB can dampen credit growth. Applying the same reasoning, all other capital buffer requirements in the Basel III Accord could be included. Unlike the other buffer requirements, however, CCyB is designed to be adjustable over time in order to counteract pro-cyclicality in the financial system. For this reason CCyB is included, and the other buffer requirements are not. Taxes and fees (e.g., regulations on tax deductible interests paid and stamp fees) is another type of instruments which could be included in the analysis. Whether to reckon taxes and fees among the macroprudential instruments or not, seems to differ quite substantially among the countries included in the sample. Lack of consistent and comprehensive data on taxes and fees necessitates omitting this type of instruments from the analysis. Information about the active use of the different measures in the countries in the panel is found in Table 1.

**Table 1.** Use of macroprudential policies in the estimation sample. The array elements indicate in which periods LTV, LTI, DSTI and Amort were used, in which periods RW was elevated, and the current level of CCyB in the various countries. Periods not entered and empty cells represent the numerical value 0.

| | LTV | LTI * | DSTI | Amort | RW | CCyB |
|---|---|---|---|---|---|---|
| Australia | | | 2017(2)-17(2) | 2017(2)-17(2) | 2004(4)-17(2) | |
| Canada | 2008(4)-17(2) | | 2008(4)-17(2) | 2008(4)-17(2) | | |
| Denmark | 2015(4)-17(2) | | 2016(1)-17(2) | | | |
| Finland | 2016(3)-17(2) | | | | | |
| Ireland | 2015(1)-17(2) | 2015(1)-17(2) | 2012(1)-17(2) | | 2007(1)-17(2) | |
| Netherlands | 2013(1)-17(2) | | 2013(1)-17(2) | | | |
| New Zealand | 2013(4)-17(2) | | | | | |
| Norway | 2010(1)-17(2) | 2010(1)-11(4) 2017(1)-17(2) | 2011(4)-17(2) | 2015(3)-17(2) | 1998(3)-01(1) 2014(1)-17(2) | 2015(3)-16(2), 1% 2016(3)-17(2), 1.5% |
| Sweden | 2010(4)-17(2) | | | 2016(2)-17(2) | 2013(2)-17(2) | 2015(4)-16(2), 1% 2016(3)-17(1), 1.5% 2017(2)-17(2), 2.0% |
| UK | | 2014(4)-17(2) | 2014(2)-17(2) | | | |

Note: *: In Norway, LTI is specifically a DTI-requirement.

Each of the indicators representing the application of the macroprudential instruments LTV, LTI, DSTI and Amort was assigned the numerical value 1 if the instrument has been implemented and activated by the public authorities in the relevant country in the relevant period of time, and 0 otherwise. Risk weights on bank's exposures (explicit or implicit) are inherent in any capital adequacy regulation. Applying the same rule for assigning numerical values to the RW indicator therefore seems little fruitful. Instead, the RW indicator was assigned the numerical value 1 if the risk weight on mortgage loans has been increased in the relevant country in the relevant period of time, and 0 if the risk weight has been lowered. In all periods that the risk weight has been kept unchanged, the RW indicator was assigned the same numerical value as in the previous period. However, the RW indicator is not calibrated to reflect the transition from one generation of Basel Accords to the next.

---

[1]   While LTI refers to the ratio between an individual loan to a household's income, DTI refers to the ratio between a household's total debt to income.

The CCyB indicator was assigned numerical value (in per cent) pursuant to the current regulations in the relevant country.

Two special cases need to be commented. In the Netherlands, the market participants established self-regulation prior to the codification into national law in 2013. In accordance with the rules outlined above, self-regulation does not qualify to be assigned the numerical value 1. Hence, LTV and DSTI are assigned the numerical value 1 for the Netherlands only as of first quarter 2013, when the code of conduct was incorporated into national law. As for Canada, macroprudential instruments were implemented and activated well in advance of 2008. The Canadian regulation was in general softened until 2007, and tightened from 2008. This could motivate several different approaches with regard to assigning indicator values. For simplicity, in this analysis, periods in which regulations have been softened were regarded as if there is no instruments activated. Accordingly, LTV, DSTI and Amort were assigned the numerical value 1 for Canada as of fourth quarter 2008.

The data on usage of macroprudential instruments were mostly collected from the International Monetary Fund (IMF), the European Systemic Risk Board (ESRB) and national authorities (financial supervisory authorities and central banks) (see Appendix B for details). To tackle contradictory information provided by the data sources, a set of guiding principles was employed. Closeness between the institution and the data, and to what extent the institution has been utilising the data, were emphasised in the assessment.

There is little doubt that the complete discretisation ("on"/"off") is an oversimplification, and that it is credible that there have been examples of gradualism (tightening and loosening of measures) in macroprudential policies. However, to ensure correct judgements about stricter and less strict policies requires expert knowledge of individual countries that we cannot claim to have. Moreover, it seems plausible that statistically and economically significant response to a change in an indicator variable is a clear sign that also a more refined operational measure of macroprudential policy change will be significant. In any case, we refer to the studies with larger cross-section dimension, a broader definition of non-interest rate policies, but also a shorter sample, for estimated effects of tightening/loosening of policies (e.g., Akinci and Olmstead-Rumsey (2017) and Richter et al. (2018)).

The house price and credit data were retrieved from the Bank of International Settlements (BIS) Statistical Warehouse (see Appendix C.1 for details). All data are seasonally unadjusted and measured on a quarterly basis.

## 4. Method

As noted, there is a new literature on econometric assessment of the effects of macroprudential policy instruments by the use of aggregate panel datasets. There are several and notable differences between the studies, including the operationalisation of the policy variables, the number of time periods and countries included in the sample and preference for estimation method.

However, there is also considerable common ground. All existing studies attempt to estimate the effects on at least two variables, credit growth and housing price change, and by the use of separately estimated equations. The explanatory variables usually belong to three categories: (i) one or more lags of the dependent variable; (ii) economic control variables (e.g., GDP and GNI growth); and (iii) a battery of policy variables. It is custom to include the monetary policy interest rate (i.e., central bank rate) as one of the control variables. This makes sense since it is easy to imagine that the monetary transmission mechanism creates a dependency between the central bank rate and the interest rate that affects housing demand, and hence the evolution of housing price and credit growth.

### 4.1. A Stylised Model Equation

A model that includes the mentioned elements, with a single macroprudential instrument for notational convenience, is:

$$y_t = \varphi_{10} + \varphi_{11} y_{t-1} + \varphi_{12} r_{t-1} + \beta_{10} pol_t + \beta_{11} pol_{t-1} + \gamma_{10} z_t + \varepsilon_{yt} \tag{1}$$

$$r_t = \varphi_{20} + \varphi_{21} y_{t-1} + \varphi_{22} r_{t-1} + \beta_{20} pol_t + \beta_{21} pol_{t-1} + \gamma_{20} z_t + \varepsilon_{rt} \tag{2}$$

where $y_t$ symbolises either credit growth or housing price change in period $t$ (measured as a growth rate or percent change), $r_t$ denotes the central bank rate, $pol_t$ denotes the macroprudential policy indicator and $z_t$ represents the mentioned control variable. In this formulation, we assume that $z_t$ is a valid conditioning variable (to save space there are no lags in this variable).

Equation (1) represents a simplified but typical model equation in the literature. Equation (2) is consistent with the universal treatment of the policy interest rate as an endogenous variable, though no studies actually estimate that equation.

Appendix A contains a derivation of Equations (1) and (2) that commences from a simultaneous equation model (SEM), where not only $y_t$ and $r_t$, but also $pol_t$ are endogenous variables. The aim of the derivation is to clarify whether the endogeneity of $pol_t$ in the system (i.e., SEM or VAR), transfers to the partial model of the system represented by Equations (1) and (2). The answer is "no", because Equations (1) and (2) can be valid conditional models, which implies that $pol_t$ is instantaneously uncorrelated with the disturbance in for example Equation (1). Hence, while $pol_t$ is endogenous in the system, it can be a predetermined variable in the two derived conditional model equations, in Equation (1) specifically, which is our main model equation of interest.

### 4.2. Parameters of Interest

The model equations that we estimated are empirical versions of Equation (1). In the literature, there is some disagreement about parameters of interest. Kuttner and Shim (2016) stated that the individual coefficients in the distributed lag on the policy variables are "of little intrinsic interest"; it is the sum of the coefficients that is of interest. In the notation above, this corresponds to having $(\beta_{10} + \beta_{11})$ as the parameter of interest. However, other studies focus on a single coefficient in the estimated models, cf. Akinci and Olmstead-Rumsey (2017) who estimated models with a single lag in each policy variable (corresponding to setting $\beta_{11}$ as the parameter of interest).

Below, we report baseline model equations for credit growth and house prices, which show point estimates of $\beta_{10}$ and $\beta_{11}$ individually. In the tables that summarise the results, we show $\beta_{10}$, the *Impact* coefficient, together with long-run coefficient, which in the case of Equation (1) is defined as:

$$\text{Long-run} = \frac{(\beta_{10} + \beta_{11})}{1 - \varphi_{11}} \tag{3}$$

noting however that, since the interest rate $r_t$ is endogenous in general, the reported long-run coefficients are only suggestive of the longer-term effects of a policy. Even in our simplified formulation, to represent the full effect, it is required that $\beta_{20} = \beta_{21} = \varphi_{21} = 0$. To assess these conditions requires a more system oriented approach than is usual in the literature, and which goes beyond the scope of the paper.

### 4.3. Estimation Methods

As noted above, the policy variable may be regarded as pre-determined in the conditional model, which corresponds logically to the model equations that we estimate, also when we start from the premise that it is endogenous in the wider economic system. Hence, both the principle of least squares and the principle of method-of-moments estimators (IV methods) give rise to consistent estimators

(in the time series dimension) if the model equation is econometrically well specified (i.e., the residuals are approximately Gaussian).

In the context of time-series cross-sectional (TSCS) and panel data, the direct parallel to OLS estimation is least square dummy variable estimation (LSDV), also called the within estimator. The finite $T$ bias of LSDV, which mainly affects the autoregressive parameter $\varphi_{11}$ but which also in principle affects the estimation of coefficients of current and lagged conditioning variables, is known as Hurwicz-bias (Hurwicz (1950)). It is an inevitable consequence of dynamics, since $y_{t-1}$ is a pre-determined variable, not a strictly exogenous variable.

However, in most cases, the magnitude of the Hurwicz-bias of $\hat{\varphi}_{11}$ is not large, and it declines quite rapidly toward zero as $T$ increases. This continues to hold for TSCS and panel data and the LSDV estimator is $T$-consistent. $\hat{\varphi}_{11}$ is asymptotically normally distributed, and the conventional $t$-statistic can be used to test hypotheses, because the t-statistic has a standard normal distribution under the null-hypothesis $\varphi_{11} = 0$. The same property holds for the t-values of the other OLS based estimators of the model equation. Importantly our main parameters of interest are $\beta_{10}$ and the long-run coefficient in Equation (3).

Hence, although the Hurvicz-bias is not alleviated by the cross-sectional dimension of the data (the LSDV estimator is not $N$-consistent), we note that our dataset contains a large number of time series observations, while there are only ten cross section units, and in such sample situations there are few (new) problems raised by estimating model equations of the type in Equation (1) by LSDV (Biorn (2017, Chp. 8.3)). Finally, as we note below, the LSDV estimator can be robustified against the influence of sample variation that the model equation has little to say about, notably structural breaks.

In the random effects model, RE, the inherent equi-correlation damages the predeterminedness of $y_{t-1}$, which may make the feasible RE estimators become both $T$ and $N$ inconsistent (see, e.g., Biorn (2017, Chp. 8.4)). In principle that problem is fixed by the Arellano and Bond GMM estimation method for dynamic macro models (see Arellano and Bond (1991)). In practice, the availability of valid and relevant instrumental variables determines what is gained by panel GMM, if the VAR is relatively well specified in the first place (see Bun and Windmeijer (2010)). It may happen that the fix for a relatively small estimation issue (lack of consistency in the cross-section direction) creates a relatively large cost of inference due to, e.g., an increase in estimated coefficient variance.

However, if we abstract from the dynamic formulation of the model equation, we note that Bell and Jones (2015) pointed out that RE estimation has other features which can make that method attractive for a dataset such as ours, when the purpose is to estimate coefficients that are related to specific effects in the countries in the sample, namely the implementation of macroprudential policies. For this reason, we include the results of RE model estimation in the section on robustness and sensitivity analysis below.

In addition to estimating the effect ($\beta_{10}$) from the introduction of the policy, and the requisite ($\beta_{10} + \beta_{11}$) for long-term effect of unchanged policy, we might be interested in estimating dynamic effects, e.g., four quarters after the introduction, two years after, and so on. These parameters, although well defined in the VAR, cannot in general be efficiently estimated from Equation (1) alone. The required condition is $\varphi_{21} = 0$ in Equation (2), i.e., that the system is characterised by one-way Granger causality.

In practice, the order of dynamics of the VAR must be specified empirically. Underspecification of lag order can imply residual mis-specification, i.e., the assumption about approximate Gaussian VAR disturbances may become untenable, and the statistical inference can become unreliable.

In addition to dynamic specification, the controls are important for securing near white-noise residuals. In this study, the controls took two forms: First, we included economic variables that are likely to be correlated with credit growth and house price changes in open economies: world stock price change, oil price change and growth in international trade. The second class of controls consists of country and period specific indicator variables for breaks in the intercept of the relationship, i.e., temporal location shifts. Because such location breaks often will be correlated with one or more of the regressors, the estimation of the parameters of interest will in general be robustified by the

inclusion of significant indicators for breaks (see Hendry and Johansen (2015), Hendry (2018), and Castle et al. (2013)).

The determination of indicators for location shifts was done objectively with the aid of the computer implemented algorithm *Impulse Indicator Saturation (IIS)* in Autometrics (cf. Hendry et al. (2008) and Hendry and Doornik (2014)), using relatively strict significance levels for one country dataset at a time. The significant indicators were then added to the panel dataset and the empirical model equations of Equation (1) were estimated, augmented by the complete set of indicators. These coefficient estimates, following the theoretical developments of Johansen and Nielsen (2009) and the TSCS and panel data application in Nymoen and Sparrman (2015), are reported with the label LSDV-IIS in the section with empirical results.

### 4.4. Nominal or Real House Price and Credit Growth?

Another important decision is the measurement of the dependent variable, $y_t$, above. In the existing literature, the custom has become to model changes in real housing prices and in real credit. However, from a regulator's point of view, the aim is certainly to affect the increase in nominal credit growth, which by itself is an argument for modelling the nominal changes directly. Moreover, as mentioned above, if the information set of the panel data estimation under-specify the data generating process of nominal price level dynamics, the decision to model real growth rates instead of nominal entails a measurement error in the dependent variable.

Interestingly, in a study by Richter et al. (2018), the research question is how macroprudential policies interact with the core objectives to stabilise prices and output. The study focuses on the effects of changes in LTV limits and finds that: "The price response appears to be slightly positive but insignificant [...] and imprecise". The implication seems to be that, unless other macroprudential instruments should happen to be more relevant for price level changes, the custom of modelling real credit and real house price responses lead to mis-measured left-hand side variables. While this type of measurement error does not necessarily bias the estimates (i.e., as long as residual autocorrelation is avoided), it is nevertheless an unnecessary complication which is likely to lead to somewhat inflated estimated coefficient standard errors.

Hence, our main results are for models where the dependent variables were nominal credit and price changes. However, we did take the custom of the literature into account by presenting sensitivity analysis where, e.g., real credit growth was the dependent variable of the estimated model equation.

## 5. Results

As noted above, the number of time periods in our sample is relatively long and covers the 1980s for most countries. This is helpful for the empirical specification of the autoregressive lag structure, which is likely to be of a relatively high order in any historical period. At the same time, it is clear that the information about macroprudential policies comes from the last part of the sample period. This raises the issue about possible sample dependency in the estimated effects of the new policies. However, comparison of the results reported by Kuttner and Shim (2016) (long time series), with the findings of Cerutti et al. (2015) and Akinci and Olmstead-Rumsey (2017) (data start in 2000) does not suggest that sample dependency is a main problem. Another indication of the same is that the authors of the existing papers do not discuss sample selection as a source of over-/underestimation of effects.

### 5.1. Credit Growth

Equation (4) shows a baseline empirical model for nominal credit growth ($\Delta c_t$), estimated by LSDV (dummies for countries), and with robust standard errors. We show the results for the autoregressive part, for the interest rate and for the six macroprudential policy variables. Note that we modelled total credit to households, and not only housing credit.

$$
\begin{aligned}
\Delta c_t \;=\; & \underset{(0.055)}{0.234}\ \Delta c_{t-1} + \underset{(0.031)}{0.205}\ \Delta c_{t-2} + \underset{(0.077)}{0.121}\ \Delta c_{t-3} + \underset{(0.068)}{0.354}\ \Delta c_{t-4} \\[4pt]
& -\ \underset{(0.029)}{0.069}\ r_{t-1} + \underset{(0.028)}{0.039}\ r_{t-2} - \underset{(0.146)}{0.464}\ LTV_t + \underset{(0.162)}{0.468}\ LTV_{t-1} \\[4pt]
& -\ \underset{(0.131)}{0.232}\ LTI_t + \underset{(0.198)}{0.3454}\ LTI_{t-1} - \underset{(0.163)}{0.007}\ DSTI_t - \underset{(0.149)}{0.089}\ DSTI_{t-1} \\[4pt]
& -\ \underset{(0.226)}{0.722}\ RW_t + \underset{(0.14)}{0.422}\ RW_{t-1} + \underset{(0.209)}{0.070}\ Amort_t - \underset{(0.202)}{0.280}\ Amort_{t-1} \\[4pt]
& +\ \underset{(0.088)}{0.231}\ CCyB_t + \underset{(0.112)}{0.036}\ CCyB_{t-1}
\end{aligned}
\tag{4}
$$

In the second line in Equation (4), the estimation results for the two lags of the central bank rate, $r_{t-1}$ and $r_{t-2}$, are included. The coefficients give estimated percentage points effects of a unit change in the interest rate. Hence, if the interest rate were increased from two to three per cent in period $t$, Equation (4) implies that nominal credit growth is expected to be reduced by 0.07 per cent in the following quarter (i.e., the point estimate of $r_{t-1}$ in the second line of Equation (4) with two decimals). This estimate is also found in the "memo" part of Table 2, in the column labelled Impact effect in the part of the table with LSDV estimation results.

**Table 2.** Credit growth. Estimated (LSDV and LSDV-IIS) effects for LTV, LTI, DSTI, RW, Amort and CCyB. Results for the central bank rate shown in the memo part of the table. t-values are shown below the estimated impact and long-run effects. Statistical significance (two sided test) is indicated by ∗∗ (5% level) and ∗ (10% level).

| | LSDV | | LSDV-IIS | |
|---|---|---|---|---|
| | Impact | Long-Run | Impact | Long-Run |
| LTV | −0.46 | 0.05 | −0.42 | 0.30 |
| | −3.18 ** | 0.06 | −3.56 ** | 0.52 |
| LTI | −0.23 | 1.31 | −0.16 | 2.11 |
| | −1.77 * | 1.32 | −1.12 | 1.98 ** |
| DSTI | −0.01 | −0.95 | 0.14 | 0.98 |
| | −0.04 | −1.56 | 1.06 | 1.02 |
| RW | −0.72 | −3.47 | −0.62 | −2.36 |
| | −3.19 ** | −1.67 * | −3.72 ** | −2.27 ** |
| Amort | 0.07 | −2.43 | −0.02 | −3.21 |
| | 0.033 | −2.14 ** | −0.14 | −2.63 ** |
| CCyB | 0.23 | 3.09 | 0.23 | 3.20 |
| | 2.62 ** | 1.85 ** | 2.31 ** | 3.00 ** |
| Memo: | | | | |
| Interest rate (1 pp increase) | −0.07 | −0.35 | −0.05 | −0.11 |
| | −2.34 ** | −2.52 ** | −2.14 ** | −1.01 |

While the estimated coefficient of $r_{t-1}$ is negative, the coefficient of $r_{t-2}$ is positive. An estimated sign-change of the interest rate effect is not uncommon in the existing studies, (see, e.g., Kuttner and Shim (2016) and Akinci and Olmstead-Rumsey (2017)). As noted above, despite the sign change, it is not a given thing that the estimated long-run effect is smaller in magnitude than the impact effect. It can go both ways, depending on the estimated sum of the autoregressive coefficients. In Table 2, we see that the long-run effect of a permanent increase in the interest rate on nominal credit growth is estimated to be −0.35 with t-value of −2.52, which is significant at the 1% level.

Turning to the macroprudential policy indicators, Equation (4) and Table 2 show several interesting results. Both LTV and LTI are estimated to have reduced nominal credit growth when they were introduced. The effects are numerically significant: reducing the quarterly growth rate by 0.46 (LTV) and 0.23 (LTI), respectively. These effects are also statistically significant, though strongly so only for LTV. For both indicators the coefficient of the lagged indicator is estimated to be positive. In Table 2, the consequence is that the two long-run coefficients are estimated to be positive, but statistically insignificant.

Equation (4) shows that debt service to income (DSTI) is estimated to have a weak effect on credit growth on impact. Although Table 2, shows that the estimated long-run effect is numerically quite large ($-0.95$), it is not statistically significant. The results for the two coefficients for the RW policy indicator are quite different, suggesting both a significant impact effect ($-0.72$ with t-value $-3.19$), as well as a numerically sizeable long-run effect, estimated to be $-3.47$ in Table 2.

The estimated impact effect of the Amort policy indicator is statistically insignificant. The long-term effect of Amort is however estimated to be negative, sizeable ($-2.43$) and statistically significant.

Finally, the countercyclical buffer variable, CCyB, is associated with positive coefficients in this estimation. Hence, there is no support for the view that due to, e.g., increased financing costs, build up of CCyB has dampened credit growth. However, we also remember that the sample information about CCyB comes from only two countries, Sweden and Norway, and one possibility is that CCyB picks up a close correlation between credit growth and introduction of Basel III capital buffer requirements. In that interpretation, where CCyB is a control variable rather than an instrument, it is interesting to note that the results for the remaining five policy indicators are in all important respects unchanged when CCyB is dropped from the empirical model.

We next turn to the results of the robust estimation, using LSDV augmented by indicators for structural breaks as explained above. The results are summarised in the columns of Table 2 labelled LSDV-IIS. The automatic IIS method found 96 indicator variables that were added to the model equation used for the LSDV estimation.

Looking at the Impact-column first, we see that many of the LSDV estimated coefficients are robust. Although there is a tendency towards smaller magnitudes, the impression that LTV, maybe LTI, and definitely RW may have had negative effects when introduced, remains. The columns with the Long-run effects show larger differences between LSDV estimation and LSDV-IIS estimation. However, the estimated coefficients of RW and Amort are numerically and statistically significant also when LSDV-IIS is used.

## 5.2. Housing Price Growth

The estimated model in Equation (5) shows a similar specification for nominal house price change ($\Delta p_t$), as we had for credit growth. The only difference, in terms of specification, is that there is a fifth autoregressive term in the house price model (it has t-value of $-4.8$).

$$
\begin{aligned}
\Delta p_t = \quad & \underset{(0.064)}{0.360}\ \Delta p_{t-1} + \underset{(0.028)}{0.012}\ \Delta p_{t-2} + \underset{(0.018)}{0.117}\ \Delta p_{t-3} + \underset{(0.053)}{0.368}\ \Delta p_{t-4} \\[4pt]
& - \underset{(0.038)}{0.1837}\ \Delta p_{t-5} - \underset{(0.073)}{0.177}\ r_{t-1} + \underset{(0.070)}{0.147}\ r_{t-2} \\[4pt]
& - \underset{(0.324}{1.04}\ \text{LTV}_t + \underset{(0.316}{0.990}\ \text{LTV}_{t-1} - \underset{(0.472)}{0.300}\ \text{LTI}_t - \underset{(0.316)}{0.04}\ \text{LTI}_{t-1} \\[4pt]
& - \underset{(0.431)}{0.027}\ \text{DSTI}_t + \underset{(0.562)}{0.268}\ \text{DSTI}_{t-1} - \underset{(0.397)}{0.811}\ \text{RW}_t + \underset{(0.494)}{0.469}\ \text{RW}_{t-1} \\[4pt]
& - \underset{(0.724)}{0.902}\ \text{Amort}_t + \underset{(0.826)}{0.833}\ \text{Amort}_{t-1} + \underset{(0.386)}{0.384}\ \text{CCyB}_t - \underset{(0.326)}{0.270}\ \text{CCyB}_{t-1}
\end{aligned}
\tag{5}
$$

In addition, in the house price change equation, the coefficient of the lagged central bank rate is estimated to be negative, and with a numerically larger coefficient than in the credit growth equation ($-0.18$). In the same way as for credit, there is a sign change for $r_{t-2}$, which is positive and with almost the same magnitude as the coefficient for $r_{t-1}$. Hence, the estimated magnitude of the long-run effect is smaller than the short-run effect ($-0.18$ and $-0.09$) in the last row of Table 3.

**Table 3.** House price change. Estimated (LSDV and LSDV-IIS) effects for LTV, LTI, DSTI, RW, Amort and CCyB. Results for the central bank rate shown in the memo part of the table. t-values are shown below the estimated impact and long-run effects. Statistical significance (two sided test) is indicated by ∗∗ (5% level) and ∗ (10% level).

| | LSDV | | LSDV-IIS | |
|---|---|---|---|---|
| | **Impact** | **Long-Run** | **Impact** | **Long-Run** |
| LTV | $-1.04$ | $-0.16$ | $-0.98$ | $-0.28$ |
| | $-3.22$ ** | $-0.48$ | $-2.88$ ** | $-0.70$ |
| LTI | $-0.30$ | $-0.79$ | $-0.18$ | $-0.72$ |
| | $-0.64$ | $-1.25$ | $-0.39$ | $-1.22$ |
| DSTI | $-0.03$ | $0.74$ | $0.08$ | $0.97$ |
| | $-0.06$ | $0.51$ | $0.19$ | $0.63$ |
| RW | $-0.811$ | $-1.05$ | $-0.79$ | $-0.81$ |
| | $-2.04$ ** | $-0.84$ | $-2.30$ ** | $-0.74$ |
| Amort | $-0.90$ | $-0.21$ | $-0.99$ | $-0.78$ |
| | $-1.25$ | $-0.19$ | $-1.37$ | $-0.64$ |
| CCyB | $0.38$ | $0.34$ | $0.46$ | $0.54$ |
| | $1.00$ | $0.28$ | $1.09$ | $0.43$ |
| Memo: | | | | |
| Interest rate (1 pp increase) | $-0.18$ | $-0.09$ | $-0.18$ | $-0.07$ |
| | $-2.43$ ** | $-2.10$ ** | $-2.73$ ** | $-1.71$ * |

In addition, for the macroprudential policy indicators, Equation (5) and Table 3 show results that are broadly consistent with what we obtained for the credit growth model. In the LSDV estimation, all measures except CCyB get negative impact coefficients, and for LTV and RW the coefficients are sizeable ($-0.9$ is the average) and statistically significant. The strong results for those two measures also carry over to the LSDV-IIS estimation.

*5.3. Tests of Joint Significance and Diagnostic Tests*

Table 4 contains additional tests based on the empirical models that we have considered thus far, i.e., for nominal credit growth and nominal house price change. First, the table shows several tests of joint statistical significance of groups of variables. We see that the dynamic augmentations of the models ("autoregressive terms") are highly significant, and so are the included controls and the policy interest rate. Interestingly, the macroprudential policy indicators are convincingly significant when the joint tests are used. That observation suggests that a certain synchronisation of policies has taken place, and that this may have increased the overall efficiency of macroprudential instrument use.

The second part of the table shows sample size and number of parameters estimated, the multiple correlation coefficient and two standard tests of residual autocorrelation ($AR(1)$ and $AR(2)$). There is indication of significant first order residual autocorrelation in Table 3 (house prices), but not in the robust estimation (moreover the autocorrelation is negative, implying that the t-values of the LSDV estimation may have been underestimated).

**Table 4.** Tests of joint significance and diagnostic tests.

| | | Table 2 | | Table 3 | |
|---|---|---|---|---|---|
| | | **LSDV** | **LSDV-IIS** | **LSDV** | **LSDV-IIS** |
| *Tests of joint significance:* | | | | | |
| Autoregressive terms | $\chi^2(4)$ | 8221 *** | 3380 *** | | |
| | $\chi^2(5)$ | | | 4362 *** | 789 *** |
| Controls | $\chi^2(6)$ | 90.3 *** | 82.1 *** | 125.4 *** | 171 *** |
| Interest rate | $\chi^2(2)$ | 10.3 *** | 4.75 ** | 8.25 *** | 9.62 *** |
| Macroprudentials | | | | | |
| -All | $\chi^2(12)$ | 992 *** | 307 *** | 145.5 *** | 664.1 *** |
| -Impact: | $\chi^2(6)$ | 20.3 *** | 44.3 *** | 62.5 *** | 109.7 *** |
| -lags: | $\chi^2(6)$ | 86.2 *** | 303.7 *** | 16.0 *** | 14.9 *** |
| *Diagnostics:* | | | | | |
| Number of observations | | 1264 | 1264 | 1589 | 1589 |
| Number of parameters: | | 34 | 130 | 35 | 119 |
| $R^2$ | | 0.68 | 0.82 | 0.39 | 0.58 |
| $AR(1)$-test | | 1.34 | 1.15 | −2.15 ** | −0.15 |
| $AR(2)$-test | | 0.04 | −0.22 | 0.97 | 0.78 |

### 5.4. Comparison with Findings in the Existing Literature

The significant results of the joint tests of the relevance of included policy instruments confirm what appears to have become the consensus view, namely that macroprudential measures can be relied upon to aid the management of the financial cycle. As can be expected, because of data and methodological differences, the results for the individual polices contain both confirmation and departures form the results in the existing empirical literature.

The result about changes in LTV ratios being associated with lower credit growth and lower house price growth sits well with the findings in the existing empirical literature. This instrument is comparable across countries, but it is nevertheless interesting that it stands out also in our sample of advanced open economies. Due to the dynamics of the model equation the implication is that the reduction of the growth rates lasts for several periods, consistent with, e.g., Kuttner and Shim (2016). However, there is no support in our results for a permanent effect on the growth rates of the credit or house price indices, only for a long-term level effect on the two target variables.

The results above also support the view that LTI, RW and Amort are relevant instruments for the management of the financial cycle. There are less consistent findings in the literature about these measures, possibly because they have been difficult to pick up in the studies that use considerably broader cross sections than we do. Although our results show that the coefficient of DSTI has the expected negative sign, there is less strong support for this instrument than in the studies by Kuttner and Shim (2016) and Cerutti et al. (2015).

### 5.5. Sensitivity Analysis

In this section, we first present how the results in Tables 2 and 3 change when the dependent variables are changed from nominal to real growth rates. The results for real credit growth are shown in Table 5. The estimated impact coefficients of LTV, LTI and RW are robust when LSDV is the estimation method. The standard errors of the coefficients are somewhat larger, which affects the magnitude of the t-values. This is consistent with the argument presented in Section 4.4 about induced measurement error. The long-run coefficients of RW and Amort in the LSDV estimation results are only partially robust when the dependent variable is changed from nominal to real credit growth: The coefficient of RW is reduced by one third in magnitude but remains significant, while Amort is reduced even more and becomes statistically insignificant.

**Table 5.** Real credit growth. Estimated (LSDV and LSDV-IIS) effects for LTV, LTI, DSTI, RW, Amort and CCyB . Results for the central bank rate shown in the memo part of the table. t-values are shown below the estimated impact and long-run effects. Statistical significance (two sided test) is indicated by ∗∗ (5% level) and ∗ (10% level).

| | LSDV | | LSDV-IIS | |
|---|---|---|---|---|
| | Impact | Long-Run | Impact | Long-Run |
| LTV | −0.63 | 0.00 | −0.55 | 0.20 |
| | −2.13 ** | 0.00 | −2.01 ** | 0.38 |
| LTI | −0.19 | 0.29 | −0.13 | 0.73 |
| | −1.07 | 0.66 | −0.71 | 1.08 |
| DSTI | −0.01 | −1.65 | 0.13 | −0.66 |
| | −0.03 | −6.02 ** | 0.90 | −1.21 |
| RW | −0.68 | −2.53 | −0.58 | −1.91 |
| | −3.11 ** | −1.99 ** | −3.72 ** | −2.76 ** |
| Amort | 0.60 | −0.60 | 0.49 | −1.31 |
| | 0.033 | −1.11 | 2.12 ** | −2.75 ** |
| CCyB | 0.00 | 1.47 | 0.01 | 1.61 |
| | 0.01 | 1.13 | 0.06 | 1.47 |
| Memo: | | | | |
| Interest rate (1 pp increase) | −0.11 | −0.22 | −0.08 | −0.08 |
| | −2.59 ** | −2.62 ** | −1.71 * | −0.93 |

The results for real house price growth are shown in Table 6. Comparison with Table 3 shows that the estimated impact coefficients of LTV and RW are significant in both tables, and hence can be seen as robust to how the left hand side variable has been measured.

**Table 6.** Real house price change. Estimated (LSDV and LSDV-IIS) effects for LTV, LTI, DSTI, RW, Amort and CCyB . Results for the central bank rate shown in the memo part of the table. t-values are shown below the estimated impact and long-run effects. Statistical significance (two sided test) is indicated by ∗∗ (5% level) and ∗ (10% level).

| | LSDV | | LSDV-IIS | |
|---|---|---|---|---|
| | Impact | Long-Run | Impact | Long-Run |
| LTV | −1.10 | −0.13 | −1.06 | −0.25 |
| | −2.16 ** | −0.34 | −2.55 ** | −0.53 |
| LTI | −0.18 | −0.46 | −0.07 | −0.41 |
| | −0.37 | −0.72 | −0.16 | −0.67 |
| DSTI | 0.02 | 0.38 | 0.13 | 0.62 |
| | 0.06 | 0.99 | 0.29 | 0.41 |
| RW | −0.76 | −0.90 | −0.71 | −0.67 |
| | −2.22 ** | −0.87 | −2.48 ** | −0.74 |
| Amort | −0.42 | −0.28 | −0.53 | −0.82 |
| | −0.43 | −0.24 | −0.54 | −0.69 |
| CCyB | 0.10 | 0.07 | 0.17 | 0.25 |
| | 0.20 | 0.06 | 0.52 | 0.23 |
| Memo: | | | | |
| Interest rate (1 pp increase) | −0.22 | −0.19 | −0.24 | −0.18 |
| | −2.86 ** | −4.40 ** | −3.46 ** | −4.81 ** |

Table 7 shows sensitivity results for the impact coefficients on credit growth along two dimensions: Nominal/real measurement of credit growth, and the use of the fixed effects (LSDV) and random effects (RE) model. As noted in Section 4.3, there are pros and cons of the two estimation methods for a dataset such as ours, where the time series dimension is relatively long, and the cross section dimension is more narrow, a so-called TSCS dataset. Theoretically, RE estimated coefficients are hampered by inconsistency (in the $T$ dimension) when we estimate a dynamic model equation. The fixed effects model and LSDV avoid that type of inconsistency. On the other hand, the sample size is always finite in practice and therefore we know that there is a Hurwicz bias in the LSDV estimation results. In such situations, the more parsimonious RE estimation might benefit from its ability to make better use of between individual country variation. As Table 7 shows, none of the estimation results in Tables 2 and 5 are in any great extent affected by changing from LSDV estimation to RE estimation. We obtained similar results for house price change, and conclude that the results obtained for the impact coefficients are robust with respect to the two types of models of individual heterogeneity.

**Table 7.** Estimated impact coefficients on nominal and real credit growth. Estimated (LSDV, LSDV-ISS, RE, and RE-IIS) effects for LTV, LTI, DSTI, RW, Amort and CCyB. Results for the central bank rate shown in the memo part of the table. t-values are shown below the estimated coefficients. Statistical significance (two sided test) is indicated by ∗∗ (5% level) and ∗ (10% level).

| | LSDV | | LSDV-IIS | | RE | | RE-IIS | |
|---|---|---|---|---|---|---|---|---|
| | Nominal | Real | Nominal | Real | Nominal | Real | Nominal | Real |
| LTV | −0.46 | −0.63 | −0.42 | −0.55 | −0.48 | −0.63 | −0.42 | −0.55 |
| | −3.18 ** | −2.13 ** | −3.56 ** | −2.01 ** | −1.30 | −1.46 | −2.40 ** | −1.51 |
| LTI | −0.23 | −0.19 | −0.16 | −0.13 | −0.20 | −0.12 | −0.16 | −0.10 |
| | −1.77 * | −1.07 | −1.12 | −0.71 | −0.43 | −0.22 | −1.12 | −1.51 |
| DSTI | −0.01 | −0.01 | 0.14 | 0.13 | −0.01 | 0.02 | 0.13 | 0.08 |
| | −0.04 | −0.03 | 1.06 | 0.90 | −0.03 | 0.04 | 1.06 | 0.20 |
| RW | −0.72 | −0.68 | −0.62 | −0.58 | −0.66 | −0.56 | −0.62 | −0.57 |
| | −3.19 ** | −3.11 ** | −3.72 ** | −3.72 ** | −1.72 * | −1.23 | −3.72 ** | −1.48 |
| Amort | 0.07 | 0.60 | −0.02 | 0.49 | 0.19 | 0.68 | −0.03 | 0.61 |
| | 0.033 | 2.07 ** | −0.14 | 2.12 ** | −0.35 | 1.05 | −0.14 | 1.14 |
| CCyB | 0.23 | 0.00 | 0.23 | 0.01 | 0.15 | −0.11 | 0.20 | −0.06 |
| | 2.62 ** | 0.01 | 2.31 ** | 0.06 | 0.24 | −0.14 | 0.42 | −0.10 |
| Memo: | | | | | | | | |
| Interest rate | −0.07 | −0.11 | −0.05 | −0.08 | −0.07 | −0.10 | −0.05 | −0.08 |
| | −2.34 ** | −2.59 ** | −2.14 ** | −1.71 * | −2.43 ** | −3.29 ** | −2.14 ** | −2.79 ** |

## 6. Conclusions

We analysed a quarterly panel dataset consisting of ten advanced open economies, seven European countries as well as Australia, New Zealand and Canada, where macroprudential policy measures have been used since the end of the last millennium. The policy instruments include caps on loan to value and income (LTV and LTI), and debt service to income (DSTI) requirements in particular, but also risk weights (RW), amortisation (Amort) and, less used, countercyclical buffer (CCyB). We estimated dynamic panel data models, which are in line with the pre-existing literature in this field, and which include the central bank rate, and controls for common nominal and real trends.

When the hypothesis of joint significance was tested, the results are convincingly significant for the battery of macroprudential policy instruments. This result applies to both credit growth and housing price change. When it comes to individual policies, we found that some of them have had more potency than others. A main finding is that the estimated short run coefficients of LTV, LTI and RW on credit growth stand out as the most significant. The sizes of these coefficients were economically as well as statistically significant.

For Amort, we were not able to pin down an impact effect. However, the long-run coefficient of total credit growth with respect to Amort was numerically and statistically significant. The policy implication is that amortisation requirements need more time to show its effect on credit growth than LTV, LTI and RW. We found only insignificant short-run and long-run coefficients for the debt service to income indicator (DSTI). As far as we know, our investigation is the first to include countercyclical capital buffer (CCyB) of the Basel III type among the set of macroprudential measures. However, CCyB was estimated to have positive coefficients, which are all statistically insignificant. Hence, any credit dampening effects resulting from the increased financing cost due to capital buffer requirements were not found in our dataset.

The results for housing price change give the same picture, with the loan to value and the risk-weighting variables as the measures with highest statistical significance.

Our findings are not in contradiction with existing econometric panel data studies that used a broader set of countries, but which also have fewer periods with (for example) active LTV-cap and LTI-cap policies. In one respect, our results are stronger about individual policy instrument effects than most of the earlier studies, since we estimated models where all policy variables were included simultaneously from the outset, instead of being added sequentially (estimating the gross effect of one variable at a time).

Methodologically, we show that the estimating equation used in existing studies can be derived as a conditional model to form a multi-equation model where the policy variable is endogenous. Econometrically, there is no requisite to lag the policy variables, in order to mitigate endogeneity problems, and this practice can instead have made it difficult to retrieve relevant policy variables.

We also demonstrate the usefulness of the methodology developed in time series econometrics, which makes use of automatically selected impulse indicators (IIS) to form a robust OLS estimator. Since the panel data LSDV estimator is a weighted sum of OLS estimators for each country, using LSDV with indicators included, is a robust panel data estimator. Empirically, we found that the estimation results of the house price and credit models were consistent, although somewhat stronger for total credit growth than for house price changes. In all main respects, the results were robust with respect to how the dependent variables were measured (nominal or real growth), and with respect to the use of fixed effects or random effects estimation.

**Author Contributions:** Conceptualization, R.N., K.P. and J.I.S.; Data curation, K.P. and J.I.S.; Formal analysis, R.N.; Methodology, R.N.; Writing—original draft, R.N., K.P. and J.I.S.; Writing—review & editing, R.N.

**Funding:** This research received no external funding.

**Acknowledgments:** We are thankful to the editors, and to four reviewers for comments that have contributed to improve the paper considerably. The views expressed are those of the authors and do not necessarily reflect those of The Financial Supervisory Authority of Norway. Thanks to Harald Johansen and Kjersti-Gro Lindquist for comments and discussion of an earlier version.

**Conflicts of Interest:** The authors declare no conflict of interest.

## Appendix A. Endogeniety of Macroprudential Policy Instruments: Consequences for Choice of Estimation Methodology

It is quite common in the literature reviewed above to mention the endogeneity of, for example, LTV as a reason for choosing to estimate by GMM (and instrumental variables estimator). However, neither the term endogenous variable nor its counterpart exogenous variable refers to precise concepts. Depending on that clarification, the need to consider GMM estimation methodology may or may not follow.

To clarify, consider the simultaneous equation model (SEM) with three endogenous variables $y_t$, $r_t$ and $pol_t$ and a single observable exogenous variable $z_t$:

$$\underbrace{\begin{pmatrix} 1 & a_{11} & a_{13} \\ a_{21} & 1 & a_{23} \\ 0 & 0 & 1 \end{pmatrix}}_{A} \begin{pmatrix} y_t \\ r_t \\ pol_t \end{pmatrix} = \underbrace{\begin{pmatrix} b_{11} & b_{12} & b_{13} \\ b_{21} & b_{22} & b_{23} \\ b_{31} & b_{32} & b_{33} \end{pmatrix}}_{B} \begin{pmatrix} y_{t-1} \\ r_{t-1} \\ pol_{t-1} \end{pmatrix} + cz_t + \epsilon_t. \tag{A1}$$

$A$ and $B$ denote the matrices with contemporaneous and lagged coefficients, while $c$ denotes the vector with the three coefficients of $z_t$.

For simplicity, we now regard $pol_t$ as a continuous variable, along with $y_t$ and $r_t$, in which case the linear in parameters formulation is more relevant. Without loss of generality, the vector with SEM disturbances is Gaussian with zero expectation (vector) and diagonal instantaneous covariance matrix.

$pol_t$ is an endogenous variable in Equation (A1), since the $B$ matrix captures that $pol_t$ is Granger caused by $y_t$ and/or $r_t$. The two zeros in the third row of the $A$ matrix captures the idea that endogenous changes in $pol$ in any given quarter is likely to be motivated by past values of credit growth and house price changes (and associated indicators of financial stability), and not by the changes in, e.g., credit growth in the same quarter (decision and implementation lags are usually large enough to make this a reasonable assumption).

The reduced form of Equation (A1) is:

$$\begin{pmatrix} y_t \\ r_t \\ pol_t \end{pmatrix} = A^{-1}B \begin{pmatrix} y_{t-1} \\ r_{t-1} \\ pol_{t-1} \end{pmatrix} + A^{-1}cz_t + A^{-1}\epsilon_t \tag{A2}$$

assuming that the inverse matrix $A^{-1}$ exists. The reduced form residual vector $v_t = A^{-1}\epsilon_t$ is Gaussian with an invertible covariance matrix (not diagonal). Equation (A2) is of course a VAR, and by conditioning on $pol_t$, we can re-parameterise the VAR as a multiple equation model consisting of two conditional equations and one marginal equation, i.e.,

$$y_t = \varphi_{11}y_{t-1} + \varphi_{12}r_{t-1} + \beta_{10}pol_t + \beta_{11}pol_{t-1} + \gamma_{10}z_t + \varepsilon_{yt} \tag{A3}$$

$$r_t = \varphi_{21}y_{t-1} + \varphi_{22}r_{t-1} + \beta_{20}pol_t + \beta_{21}pol_{t-1} + \gamma_{20}z_t + \varepsilon_{rt} \tag{A4}$$

$$pol_t = \phi_{21}y_{t-1} + \phi_{22}r_{t-1} + \phi_{33}pol_{t-1} + \alpha_{20}z_t + v_{polt} \tag{A5}$$

where Equations (A3) and (A4) are the conditional model equations, and Equation (A5) is the marginal model equation (i.e., identical to the third row in Equation (A2)). We see that Equations (A3) and (A4) are identical to Equations (1) and (2) in the main text. They also have the same interpretation, implying that $Cov(\varepsilon_{yt}, v_{polt}) = Cov(\varepsilon_{rt}, v_{polt}) = 0$ as well as $Cov(pol_t, \varepsilon_{yt}) = Cov(pol_t, \varepsilon_{rt}) = 0$. Hence, there is no endogeneity problem due to correlation between $pol_t$ and the disturbance in the model in Equation (1). It follows that IV (or GMM) estimation of Equation (1) should be motivated by other arguments, e.g., the idea that the finite sample bias of OLS estimators for the coefficients in Equation (1) might be alleviated by GMM, which was the original motivation of Arellano and Bond (1991). This estimator addresses the second order issue of finite sample bias of dynamic panel models (see Arellano (2003, Chp. 6.3) and Baltagi (2010, Chp. 8)). However, as mentioned in the main text, empirical models of credit growth and house price changes are likely to include several lags of $y$ from the outset. For model equations of this type, it is unclear how even longer lags of $y$ (than already included) will function as GMM instrumental variables.

**Appendix B. Data Sources**

*Appendix B.1. Housing Prices and Credit*

All data are seasonally unadjusted and are measured on a quarterly basis. The data source was the Bank for International Settlements (BIS) Statistical Warehouse.

Housing price indexes for all countries were retrieved from the BIS Residential Property Price database, and are based on national sources. Further information on the housing prices series are found in https://www.bis.org/statistics/pp_long_documentation.pdf.

Household credit data for all countries were retrieved from the BIS Long Series on Total Credit. The series are based on national sources, and are compiled by data from financial accounts and the balance sheet of domestic banks. Further information can be found in https://www.bis.org/statistics/totcredit/credpriv_doc.pdf.

*Appendix B.2. Macroprudential Policies in the Sample*

The main sources of the dataset were the European Systemic Risk Board (ESRB), Overview of national macroprudential measures (which can be downloaded from: https://www.esrb.europa.eu/national_policy/html/index.en.html), various reports and publications from the International Monetary Fund (IMF) and national authorities (financial supervisory authorities and central banks). A more detailed note (in Norwegian only) outlining the groundwork for the dataset can be provided on request.

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
