# Peer review of "Estimation of Effects of Recent Macroprudential Policies in a Sample of Advanced Open Economies"

_ijfs, doi:10.3390/ijfs7020023_

Round 1
Reviewer 1 Report
Review for “Estimation of effects of recent macroprudential policies in a sample of advanced open economies”
The paper is well written. The econometrics look sound. The topic is of interest and is a timely topic.
I see two issues that need to be improved:
1. the paper is not targeted enough
2. the review of the literature is limited.
3. a graph (or two) could help the reader).
For 1, I would suggest finding a more “interesting” title and having a more focused abstract and conclusion.
For 2, please discuss briefly the issues that are discussed in the paper (credit, house prices etc). The literature in these topics is rich and you can mention some studies.
References
Chortareas, G., Magkonis, G., Moschos, D., & Panagiotidis, T. (2015). Financial development and economic activity in advanced and developing open economies: Evidence from panel cointegration. Review of Development Economics, 19(1), 163-177.
Author Response
We would like to thank the reviewer for reading our work, and for the questions as well as for the comments in the report. Please find enclosed a point-by-point answering in a separate file.

Reviewer 2 Report
The paper itself covers interesting and hotly debated problems important for policy decision-makers, and focuses on the effects of macroprudential policies on the loan and hous prices growth. Thus it fits within the wider stream of economic literature of the role of regulatory policy in economic growth. However, I do have some concerns with the paper which should be addressed by the Author(s).
Major comments:
1. The literature review on the effects on macroprudential policy on credit growth should be broadened. Definitely the paper should include one separate section for such a review, covering studies using both aggregated data as well as individual data.
2. There are no clear hypotheses presented in the text. They should be included in the literature review.
3. Looking at the current version - I can't find what is the clear added value of the research presented in this paper for the policy makers. And, what is the contribution to the literature -which should be presented in the introduction.
4. Section 3.4 (p. 6) discusses the use of nominal versus real growth rates (for credit and house prices). However, it seems reasonable to include analysis for real growth rates in the robustness checks section (which is missing in the current version of paper).
5. The paper should consider application of alternate estimators (at least for the results presented in Tables 2 and 3) - to test the sensitivity of results to the estimation method. What would be your results if you apply e.g. RE / Robust RE estimator (which could be better considering dataset presented in the study, see e.g. Bell, A., & Jones, K. (2015). Explaining Fixed Effects: Random Effects Modeling of Time-Series Cross-Sectional and Panel Data. Political Science Research and Methods, 3(01), 133–153. https://doi.org/10.1017/psrm.2014.7)
6. The results presented in should be interpreted in the context of previous research.
Minor issues:
1. There are some language mistakes, mostly grammar mistakes (e.g.: page In Canada, macroprudential instruments was implemented and activated…; page 4: extent the institution has utilizing the data…). Thus I suggest that the text sholud go through language correction.

Author Response

(The authors gave the same response as above.)

Reviewer 3 Report
Good piece of work.
Author Response
We would like to thank the reviewer for reading our work, and for the encouragement given.
Reviewer 4 Report
I hope that my comemnts will be of some help for authors to improve the quality of the article.
1. There are many papers dealing with this issue therefore the novelty of this paper needs to be highlighted.
2. The literature review should be extended.
3. The methodolgy needs more solid background.
4. The results should be discussed and compared with other studies.
5. Conclusions should be improved. Policy implications should be added.
Author Response

(The authors gave the same response as above.)

Round 2
Reviewer 4 Report
I think that the article is well revised. I am glad to say that the article can be accepted for publication.